# The Effect of PGC-1alpha-SIRT3 Pathway Activation on *Pseudomonas aeruginosa* Infection

**DOI:** 10.3390/pathogens11020116

**Published:** 2022-01-19

**Authors:** Nicholas M. Maurice, Brahmchetna Bedi, Zhihong Yuan, Kuo-Chuan Lin, Joanna B. Goldberg, C. Michael Hart, Kristina L. Bailey, Ruxana T. Sadikot

**Affiliations:** 1Department of Medicine, Division of Pulmonary, Allergy, Critical Care, and Sleep Medicine, Emory University School of Medicine, Atlanta, GA 30322, USA; nicholas.michael.maurice@emory.edu (N.M.M.); bbedi@emory.edu (B.B.); channing.lin@alumni.emory.edu (K.-C.L.); Michael.Hart3@va.gov (C.M.H.); 2Atlanta Veterans Affairs Health Care System, Decatur, GA 30033, USA; 3VA Nebraska Western Iowa Health Care System, Omaha, NE 68105, USA; zhyuan@unmc.edu (Z.Y.); kbailey@unmc.edu (K.L.B.); 4Division of Pulmonary, Critical Care & Sleep, Department of Internal Medicine, University of Nebraska Medical Center, Omaha, NE 68198, USA; 5Department of Pediatrics, Division of Pulmonology, Allergy/Immunology, Cystic Fibrosis, and Sleep, Emory University, Atlanta, GA 30322, USA; joanna.goldberg@emory.edu; 6Children’s Healthcare of Atlanta, Center for CF and Airways Disease Research, Atlanta, GA 30322, USA

**Keywords:** *P. aeruginosa*, inflammasome, peroxisome proliferator-activated receptor-γ coactivator-1α, SIRT3

## Abstract

The innate immune response to *P. aeruginosa* pulmonary infections relies on a network of pattern recognition receptors, including intracellular inflammasome complexes, which can recognize both pathogen- and host-derived signals and subsequently promote downstream inflammatory signaling. Current evidence suggests that the inflammasome does not contribute to bacterial clearance and, in fact, that dysregulated inflammasome activation is harmful in acute and chronic *P. aeruginosa* lung infection. Given the role of mitochondrial damage signals in recruiting inflammasome signaling, we investigated whether mitochondrial-targeted therapies could attenuate inflammasome signaling in response to *P. aeruginosa* and decrease pathogenicity of infection. In particular, we investigated the small molecule, ZLN005, which transcriptionally activates peroxisome proliferator-activated receptor-γ coactivator-1α (PGC-1α), a master regulator of mitochondrial biogenesis, antioxidant defense, and cellular respiration. We demonstrate that *P. aeruginosa* infection promotes the expression of inflammasome components and attenuates several components of mitochondrial repair pathways in vitro in lung epithelial cells and in vivo in an acute pneumonia model. ZLN005 activates PGC-1α and its downstream effector, Sirtuin 3 (SIRT3), a mitochondrial-localized deacetylase important for cellular metabolic processes and for reactive oxygen species homeostasis. ZLN005 also attenuates inflammasome signaling induced by *P. aeruginosa* in bronchial epithelial cells and this action is dependent on ZLN005 activation of SIRT3. ZLN005 treatment reduces epithelial-barrier dysfunction caused by *P. aeruginosa* and decreases pathogenicity in an in vivo pneumonia model. Therapies that activate the PGC-1α—SIRT3 axis may provide a complementary approach in the treatment of *P. aeruginosa* infection.

## 1. Introduction

*Pseudomonas aeruginosa* is an opportunistic bacterial pathogen responsible for a variety of acute infections, including pneumonia, sepsis, urinary tract infections, and wound infections. *P. aeruginosa* is a life-threatening cause of nosocomial and ventilator-associated pneumonia (VAP) among those in the intensive care unit and confers a mortality rate as high as 50%, even with antibiotic treatment. *P. aeruginosa* also poses a risk to immunosuppressed patients, elderly nursing home residents, and those with severe chronic obstructive pulmonary disease (COPD). In addition, *P. aeruginosa* is an important cause of chronic and recurrent respiratory infections among patients with cystic fibrosis (CF) and non-CF bronchiectasis [1]. The increase in antibiotic resistance among *P. aeruginosa* isolates is a growing global health threat. Therefore, there is great interest in developing novel therapeutic strategies that not only kill or inhibit the growth of bacteria, but also enhance the host immune response to infection [2].

The innate immune response to bacterial infection relies on a network of germline-encoded receptors called pattern recognition receptors (PRRs), which include membrane-bound Toll-like receptors (TLR) and intracellular receptors, including nucleotide oligomerization domain (NOD)-like receptors (NLRs) and AIM2 (absent in melanoma 2)-like receptors. The cytosolic receptors can recognize conserved microbial motifs called pathogen-associated molecular patterns (PAMPs) or host-derived damage-associated molecular patterns (DAMPs) and subsequently recruit the assembly of the inflammasome complex [3,4]. Oligomerization of the inflammasome complex can result in autoproteolysis of caspase-1 into an active form that subsequently activates the inflammatory cytokines, IL-1β and IL-18 [5].

*P. aeruginosa* expresses several PAMPs that are recognized by innate immune receptors. Lipopolysaccharide (LPS) can induce both TLR4-dependent signaling [6] and also noncanonical inflammasome signaling [7]. Flagellin can induce inflammatory pathways through interactions with both TLR5 and the NLRC4 inflammasome [8,9,10]. The type III secretion system apparatus of *P. aeruginosa* can also induce NLRC4 activation [8,11]. Type IV bacterial pilus can activate noncanonical inflammasome signaling. In addition, bacterial-released outer membrane vesicles have been shown to activate the inflammasome through delivered LPS and flagellin [12,13]. Furthermore, LL-37, an epithelial antimicrobial immunomodulatory peptide, can act on other epithelial cells to promote the NLRP3 inflammasome in response to *P. aeruginosa* infection [14]. Mitochondrial damage has also been shown to be critical for inflammasome signaling in response to *P. aeruginosa* infection. Mitochondrial DAMPs, such as mitochondrial DNA (mtDNA) and mitochondrial reactive oxygen species (mtROS), regulate the activation of the NLRC4 and NLRP3 inflammasome pathways [15,16,17].

Several studies have not found the inflammasome to be beneficial in contributing to the host response in acute and chronic *P. aeruginosa* lung infection. Instead, these studies suggest that inflammasome activation is harmful possibly by causing a dysregulated inflammatory response [18]. The deletion of inflammasome components, such as NLRC4, caspase 1, and caspase 11, and components of IL-1β and IL-18 signaling is protective in models of *P. aeruginosa* pneumonia [18,19,20,21]. Additionally, the inhibition of NLRP3 [22] and IL-1R [23] ameliorated airway injury in models of chronic CF-associated *P. aeruginosa* infection.

We and others have previously demonstrated that *P. aeruginosa* can induce mitochondrial damage [16,17,23,24]. Specifically, we found that the bacterial quorum-sensing molecule, *N*-(3-Oxododecanoyl)-l-homoserine lactone (3-oxo-C12-HSL), can attenuate mitochondrial bioenergetics, mitochondrial repair pathways, and induce mtROS generation and also attenuate expression of peroxisome proliferator-activated receptor-γ coactivator-1α (PGC-1α), a master regulator of mitochondrial biogenesis, antioxidant defense, and cellular respiration. ZLN005 is a small molecule that was identified in a high-throughput-screening assay to transcriptionally regulate PGC-1α expression [25]. We hypothesized that targeted transcriptional upregulation of PGC-1α with ZLN005 would attenuate inflammasome signaling and enhance the epithelial host response to *P. aeruginosa* infection.

## 2. Results

### 2.1. P. aeruginosa Induces Expression of NLRP3 Inflammasome in Bronchial Epithelial Cells

Previous studies have demonstrated that *P. aeruginosa* infection can induce inflammasome signaling in both myeloid and epithelial cells [4,26]. We investigated whether infection with the *P. aeruginosa* strain, PAO1, induced expression of components of the NLRP3 inflammasome and found that a 6-h infection significantly increased expression of NLRP3, the adapter protein ASC, caspase-1, IL-1β, and IL-18 in BEAS-2B bronchial epithelial cells (Figure 1A,B). In addition, expression of absent in melanoma (AIM)2, a distinct inflammasome receptor, was upregulated by *P. aeruginosa* infection (Figure 1A). Infection with PAO1 also significantly increased protein levels of cleaved IL-1β, demonstrating increased activation of the inflammasome pathway (Figure 1B). Upregulation of NLRP3, IL-1β, and IL-18 was also demonstrated in primary isolated human bronchial epithelial cells after a 6-h infection with PAO1 (Figure 1C). Male C57BL/6 mice were infected with PAO1 intranasally, and the left lung was homogenized and the measurement of the relative expression of IL-1β showed significant increase, compared to the untreated control (Figure 1D).

### 2.2. P. aeruginosa Attenuates Expression of Genes Involved in Mitochondrial Repair and Quality Control

Previous studies demonstrate that mitochondrial damage is a critical step in the activation of the inflammasome in response to *P. aeruginosa.* Release of mtDAMPs, such as mtROS, succinate, and mtDNA, and calcium from damaged mitochondria can promote the expression of IL-1β and downstream inflammasome processing [27]. We have previously shown that *P. aeruginosa* can induce mtROS production and oxidative mtDNA damage [24]. Mitochondrial quality control pathways are important for the repair or selective destruction of mitochondria, the enhancement of antioxidant defenses, and the creation of new mitochondria through mitochondrial biogenesis [28]. Therefore, we examined the effect of *P. aeruginosa* infection on the relative expression of several genes important for mitochondrial quality control both in vitro and in vivo. An MOI of one was chosen based on optimization experiments that found high toxicity with higher doses (MOI 10–30). Infection with the PAO1 strain in BEAS-2B bronchial epithelial cells decreases the expression of PGC-1α, mitochondrial transcription factor A (TFAM), the mitochondrial-localized NAD+ deacylase sirtuin 3 (SIRT3), and nuclear factor erythroid 2-related factor 2 (NRF2) (Figure 2A). There was no significant change in sirtuin 1 or NRF1 expression. Similarly, *P. aeruginosa* infection attenuated expression of PGC-1α and SIRT3 in primary human bronchial epithelial cells (Figure 2B). Next, we investigated whether *P. aeruginosa* infection resulted in downregulation of important mitochondrial quality-control genes in lung homogenates from an in vivo pneumonia model. Indeed, there was decreased expression of TFAM, NRF1, NRF2, SIRT1, and SIRT3 in the lungs of mice infected with PAO1 for 24 h as compared to control animals (Figure 2C) Interestingly, there was no significant change in PGC-1α at 24 h, but there was downregulation in the PGC-1α target genes, TFAM, NRF1, NRF2, and SIRT3.

### 2.3. MiR-23a Acts to Post-Transcriptionally Regulate PGC-1α in P. aeruginosa Infection

PGC-1α acts as a metabolic sensor that enables cells to respond to a variety of stimuli, including alterations in metabolic substrate availability. PGC-1α acts as a transcriptional coactivator to stimulate cellular processes, such as mitochondrial biogenesis, oxidative phosphorylation, fatty acid oxidation, and reactive oxygen species detoxification. Its activity is regulated by a variety of transcriptional, post-transcriptional, and post-translational mechanisms. Transcriptional control of PGC-1α is mediated by various regulators, including cAMP response element-binding protein (CREB), nuclear factor of activated T-cells (NFAT), myocyte enhancer factor 2 (MEF2), Yin Yang 1 (YY1), PPARs, and SIRT1. Post-translational regulation of PGC-1α has been shown to be mediated by phosphorylation, acetylation, and methylation. Finally, post-transcriptional regulation of PGC-1α by microRNA (miRNAs) have been demonstrated with miR-23a, miR-22, miR-696, miR-761, miR-199a/214, miR-29b, miR-29b, miR-19b/221/222, miR-485-3p, and miR-485-5p implicated in various organs and model systems [29]. miR-23a, in particular, has been shown to directly downregulate PGC-1α expression via binding to its mRNA 3′UTR region [30,31,32,33]. We first investigated if *P. aeruginosa* infection modulated miR-23a expression in vitro and in vivo. In bronchial epithelial cells (BEAS-2B), the 6-h infection with *P. aeruginosa* or 6-h treatment with the pseudomonal quorum-sensing molecule, *N*-3-oxo-dodecanoyl-l-Homoserine lactone (3-oxo-C12-HSL), induced the expression of miR-23a (Figure 3A). Similarly, in our mouse pneumonia model, *P. aeruginosa* infection resulted in upregulation of miR-23a in lung homogenates compared to untreated control mice (Figure 3B). Next, to determine whether miR-23a directly attenuated PGC-1α expression in bronchial epithelial cells, we treated BEAS-2B cells with a miR-23a mimic (Figure 3C) or anti-miR-23a (Figure 3D) to demonstrate that miR-23a directly attenuates basal PGC-1α expression.

### 2.4. ZLN005 Rescues Expression of the PGC-1α-SIRT3 Axis Attenuated by P. aeruginosa Infection

Previously, we demonstrated that small molecules targeting PGC-1α post-translational regulation rescue mitochondrial and metabolic derangements caused by *P. aeruginosa* infection [24]. These therapies, however, can have many off-target and pleiotropic effects. To better determine the role of PGC-1α and downstream signaling in the host response to *P. aeruginosa* infection, we investigated a small-molecule therapeutic called ZLN005 that was identified to be a specific and selective transcriptional inducer of PGC-1α expression [25]. We first confirmed that ZLN005 treatment at doses of 1 and 5 μM significantly increased expression of PGC-1α in bronchial epithelial cells, BEAS-2B (Figure 4A). ZLN005 also upregulates the expression of SIRT3, a downstream target induced by the transcriptional coactivator PGC-1α (Figure 4A). SIRT3 is a mitochondrial-localized deacetylase important for ROS detoxification and proper mitochondrial function [34,35]. Next, we investigated if ZLN005 could prevent the downregulation of PGC-1α and SIRT3 in epithelial cells after *P. aeruginosa* infection. Pretreatment with ZLN005 prevented the downregulation of PGC-1α and SIRT3 by *P. aeruginosa* in lung epithelial cells (Figure 4B,C).

### 2.5. ZLN005 Abrogates Inflammasome Signaling Induced by P. aeruginosa through Activation of the PGC-1α-SIRT3 Axis

Given the role of the PGC-1α-SIRT3 axis in preventing mitochondrial damage, which is a known trigger for inflammasome signaling, we hypothesized that activation of the PGC-1α-SIRT3 pathway might abrogate inflammasome signaling in response to *P. aeruginosa* infection. We pretreated BEAS-2B cells with ZLN005 prior to infection with PAO1 and measured expression levels in inflammasome components. Interestingly, ZLN005 treatment prevented the upregulation of NLRP3, IL-1β, and AIM2 (Figure 5A–C). To determine whether the action of ZLN005 was dependent on SIRT3, we transfected cells with either control-scrambled siRNA or siRNA directed against SIRT3 (siSIRT3) prior to treatment with ZLN005 or vehicle control with or without PAO1. As expected, siSIRT3 prevented the upregulation of SIRT3 caused by ZLN005 (Figure 6A). ZLN005 prevented the upregulation of IL-1β caused by PAO1, but this effect was abrogated in cells deficient in SIRT3 (Figure 6B). These data suggest that SIRT3 is essential for ZLN005 activity in preventing inflammasome activation by *P. aeruginosa*.

### 2.6. ZLN005 Improves Epithelial Barrier Function and Reduces Bacterial Load In Vivo

We next wanted to interrogate if ZLN005 improved the functional host response to *P. aeruginosa* infection. First, we determined whether ZLN005 treatment prevented bacterial transmigration across the epithelial barrier. Calu-3 cells grown on transwell supports were infected with PAO1 within the apical chamber. Bacterial transmigration into the basolateral chamber was measured at 6 h. ZLN005 treatment significantly attenuated bacterial transmigration (Figure 7A). Next, we tested whether or not ZLN005 treatment was protective in an in vivo pneumonia model. Mice were pretreated with ZLN005 or vehicle control by oral gavage prior to PAO1 infection. ZLN005 significantly attenuated bacterial load in the lung (Figure 7B) and in the bronchoalveolar lavage (Figure 7C). Similar to what was demonstrated in vitro, ZLN005 therapy also significantly restored expression of SIRT3 in mice infected with PAO1 (Figure 7D) and attenuated IL-1β expression in lung homogenates (Figure 7E). Collectively, these data show that ZLN005 activation of the PGC-1α-SIRT3 axis attenuated the pathogenicity of *P. aeruginosa* infection in vitro and in vivo.

## 3. Discussion

In this study, we uncovered an important pathway connecting bacterial-induced mitochondrial derangements with inflammasome-signaling and the host response to *P. aeruginosa* pulmonary infection. *P. aeruginosa* infection induces inflammasome signaling in lung epithelial cells and attenuates several important regulators of mitochondrial function including the transcriptional coactivator, PGC-1α, and its downstream mitochondrially localized target, SIRT3. The attenuation of PGC-1α is due in part to post-transcriptional inhibition via miR-23a. ZLN005, a small molecule activator that transcriptionally activates PGC-1α, abrogates inflammasome signaling induced by *P. aeruginosa* through activation of the PGC-1α-SIRT3 axis. Further, ZLN005 attenuates epithelial-barrier dysfunction caused by *P. aeruginosa*. In an in vivo acute pneumonia model, ZLN005 attenuates inflammasome signaling and attenuates the bacterial burden (Figure 7F). These data provide a rationale for targeting the PGC-1α-SIRT3 axis to improve the host response to *P. aeruginosa* lung infections.

Inflammation is critical for responding to infections and clearing tissue injury, but dysregulated or excessive inflammation can also contribute to tissue injury [36]. Innate immune-inflammatory responses depend on a network of cell-surface, endosomal, and cytoplasmic receptors that recognize pathogen- or host-derived signals. This network includes inflammasome complexes of the NLR and AIM2-like families that signal through enzymatic maturation of caspase-1, leading to IL-1β and IL-18 release [37]. Full inflammasome activation requires both transcriptional priming of NLR and IL-1β expression along with post-translational processing. Priming of the inflammasome through transcriptional upregulation of inflammasome components, mediated by proinflammatory signals acting through NF-κB, is a limiting factor in the full activation of the inflammasome [38]. We demonstrate that priming of the inflammasome activation by *P. aeruginosa* infection is attenuated by therapies targeting the PGC-1α-SIRT3 axis, critical for mitochondrial function.

Several studies have investigated the role of the inflammasome in *P. aeruginosa* infection and have not found that inflammasome activation promotes *P. aeruginosa* clearance [27]. In fact, these studies demonstrate that the deletion or inhibition of inflammasome components was protective in models of *P. aeruginosa* pneumonia [18,19,20,21,22,23]. Our present study similarly found that inhibition of the inflammasome pathway was associated with improved bacterial clearance.

Mitochondrial dysfunction is recognized as an important mediator of inflammasome signaling [39]. Mitochondrial ROS can mediate priming of inflammasome signaling pathways [40,41]. Further, mitochondrial DAMPs, such as mtDNA, mtROS, calcium, and cardiolipin, released from damaged mitochondria function to activate the inflammasome complexes [39]. Several mitochondrial quality-control pathways, including mitophagy, mitochondrial biogenesis, mitochondrial fission and fusion, and antioxidant defenses have been shown to regulate inflammasome activation in various disease states [42,43,44,45]. PGC-1α, which acts as a transcriptional coactivator to promote mitochondrial biogenesis, oxidative phosphorylation, fatty-acid oxidation, and antioxidant defense, has been shown to regulate inflammasome signaling in different model systems [46,47,48,49,50,51]. For example, in an alcoholic liver-disease model, oroxylin A, a phytochemical, was found to suppress inflammasome activation through PGC-1α [47]. However, the role of PGC-1α in modulating inflammasome activation in response to infection has not been well studied.

SIRT3 is one of seven mammalian sirtuins, which are a conserved class III subfamily of histone deacetylases (HDACs) that have NAD^+^-dependent deacetylase activity. SIRT3 is known to localize to the mitochondria where it regulates several cellular metabolic processes [47,52]. It also promotes antioxidant activity through activation of superoxide dismutase 2 and the glutathione system [53]. Regulation of SIRT3 expression is mediated by PGC-1α [34,35]. SIRT3 has a protective role in various inflammatory and fibrotic processes, including kidney injury, cardiomyopathy, and pulmonary fibrosis [51,54,55], and regulates inflammasome activation in various model systems [56,57,58,59,60,61,62,63,64,65,66]. However, there is a lack of research investigating SIRT3′s role in inflammasome activation in the host response to infection. One previous study examined the role of SIRT3 in various infection models, namely, endotoxemia, *Escherichia coli* peritonitis, *Klebsiella pneumoniae* pneumonia, listeriosis, and candidiasis. Using SIRT3 knockout (KO) mice, they found that SIRT3 deficiency was not associated with any difference in cytokine (IL-6, TNF-α) production, bacterial burden, or survival [67]. Although our study did not directly assess whether SIRT3 deficiency affected *P. aeruginosa* pneumonia severity, it did suggest that ZLN005 activation of the PGC-1α-SIRT3 axis improved bacterial clearance. These potentially conflicting results could be explained in a few ways. First, it is possible that SIRT3 KO mice are able to compensate for the loss of SIRT3 over time by recruiting other mitochondrial pathways. Second, it is possible that SIRT3 basilar expression is attenuated by various infections (as we have shown it is during *P. aeruginosa* pneumonia) and therefore, unable to contribute significantly to the host response. Finally, it is possible that ZLN005 acts through additional pathways other than the PGC-1α-SIRT3 axis to augment host response.

Our current study is the first to identify a potential role of the PGC-1α-SIRT3 axis in the regulation of the inflammasome response in bacterial infection. It is also the first to investigate the novel small molecule, ZLN005, in a model of bacterial infection and the first to investigate its role in the activation of SIRT3. Our current study also has several limitations. First, we focused on the priming of the inflammasome response rather than the proteolytic activation of the inflammasome, although both are important aspects of the inflammasome activation. Second, it is impossible to exclude that ZLN005 is exerting off-target effects independent of the PGC-1α—SIRT3 pathway. However, we do show that its effect on inflammasome suppression is dependent on SIRT3. Further, the mechanism by which PGC-1α—SIRT3 activation attenuates the inflammasome activation has not been fully elucidated. Future studies will examine whether mtROS detoxification, protection of mtDNA integrity, maintenance of mitochondrial bioenergetics, or other mechanisms are responsible for this effect. Finally, this study uses only the PAO1 strain of *P. aeruginosa.* Although PAO1 is a standard and well-studied laboratory strain with intermediate toxicity, it is neither a clinically problematic mucoid strain nor an exoU-positive strain with high cytotoxicity. Therefore, it is possible that the results may not be generalizable to all strains of *P. aeruginosa*.

In conclusion, these studies provide a rationale for a new therapeutic avenue for adjunctive therapies in refractory *P. aeruginosa* infection. *P. aeruginosa* is responsible for chronic recalcitrant infections in patients with cystic fibrosis and non-CF bronchiectasis and acute infections among hospitalized or immunocompromised patients [1,2]. Under these conditions, unregulated inflammation can lead to further tissue injury and/or progressive sepsis and multiorgan failure. Despite this, proper antimicrobial therapy, the persistence of PAMPs from degraded microbial components, and DAMPs from cellular debris can drive ongoing inflammation. Our data suggest that therapies targeting the PGC-1α-SIRT3 mitochondrial pathway can attenuate potentially damaging inflammasome activation and augment the host response to infection. This pathway could be utilized as a useful adjunctive therapy to treat *P. aeruginosa* infections and other pathogens that evade immune defenses by similar mechanisms.

## 4. Materials and Methods

### 4.1. Cell Line Models

BEAS-2B bronchial epithelial cells (ATCC, Rockville, MD, USA) were maintained with complete BEGM media (Lonza, Basel, Switzerland) supplemented with 10% FBS and plated on dishes coated with collagen-fibronectin-BSA mixture. Cells were treated with 3-oxo-C12-HSL (Sigma-Aldrich, St. Louis, MO, USA), ZLN005 (Cayman Chemical, Ann Arbor, MI, USA), or cocultured with the *P. aeruginosa* strain, PAO1, (MOI of one), for 16 h. Calu-3 cells were grown in Eagle’s minimum essential medium (EMEM, ATCC) on 3 μM polyester membrane Costar Transwell^®^ permeable supports (Corning, Waltham, MA, USA), and tight junctions were matured after a period of 7–10 days as described previously [24].

### 4.2. Normal Human Bronchial Epithelial Cells

Primary, normal human bronchial epithelial cells (NHBEs) were isolated from deidentified human lungs that were not used for transplantation. We excluded donors with a history of any lung disease, current smoking, ≥20-pack-year history of smoking, and heavy alcohol use. The protocol was approved by the LiveOn Nebraska ethics committees and the University of Nebraska Medical Center Institutional Review Board. Airway epithelial cells were isolated using a method previously described [68]. Briefly, the large airways were dissected out and protease digested. After 36–48 h, the airway lumens were scraped, and the resulting cells were plated on collagen-coated plates in bronchial epithelial growth medium (Lonza, Basel, Switzerland). Only cells with passage numbers less than five were used in experiments.

### 4.3. Bacterial Stocks

Stock of *P. aeruginosa* strain PAO1 was prepared as previously described [69]. Bacteria were used at a multiplicity of infection (MOI) of one for 16-h in vitro experiments unless otherwise stated. Cultures were adjusted to an OD600 (optical density) of 0.2 (∼1.86 × 10^9^ colony-forming units/mL) [70].

### 4.4. In Vivo Pneumonia Model

C57BL/6J mice were purchased from The Jackson Laboratory (Bar Harbor, ME, USA). All mice were housed under specific pathogen-free conditions at the Veterans Affairs Medical Center (VAMC; Decatur, GA, USA). All experiments were approved by the VAMC Institutional Animal Care and Use Committee. Male C57BL/six mice, 8 to 12 weeks of age, were treated with ZLN005 dissolved in methylcellulose (United States Biological, Salem, MA, USA) at 15 mg/kg body weight or with an equivalent volume of methylcellulose alone used as a vehicle control by oral gavage for three doses every 8 h. After 24 h, the mice were infected with PAO1 intranasally with 10^8^ CFU as described previously [70,71]. Briefly, PAO1 stocks were grown as described above and adjusted to an inoculum dose of 10^8^ CFU/50 μL. The inoculum was placed just above the nostril with a sterile pipette tip. Mice were anesthetized for the oral gavage and intranasal treatments with isoflurane inhalation. Twenty-four hours following infection, all the mice were sacrificed, and the lungs were harvested. Bronchoalveolar lavage (BAL) was performed after sacrificing mice 24 h postinfection, as described previously [71]. Briefly, mouse tracheas were exposed through a small skin incision on the anterior neck. Once exposed, a 21-gauge lavage needle was inserted into the trachea. Each mouse was lavaged three times with 1 mL of PBS. The BAL fluid was then spun at 300× *g* for 5 min at 4 °C. The supernatant was collected and stored at −80 °C. The cellular fraction was lysed using 0.5% Triton X-100. Lysates were titrated and plated in a volume of 50 μL on blood agar plates with 5% sheep blood. CFU counting was performed after 24 h. After BAL, the right lung was collected and used for bacterial colony counts. Each entire right lung was gently homogenized in 1 mL of PBS, the lysates were titrated in PBS, and 50-μL aliquots were plated on blood agar plates with 5% sheep blood (BD Biosciences). The plates were incubated overnight at 37 °C, and the colonies were counted. The colony counts were normalized to the wet weight of the lung tissue. Quantitative PCR was performed on the left lung homogenates harvested from these mice.

### 4.5. Quantitative RT-PCR

RNA was isolated using an RNeasy kit (Qiagen, Hilden, Germany) or mirVana miRNA isolation kit (Life Technologies, Carlsbad, CA, USA). RNA (1 μg) was used to synthesize cDNA using SuperScript II RT (Invitrogen, Waltham, MA, USA). Quantitative real-time PCR was performed using SYBR green probes (Applied Biosystems, Waltham, MA, USA) normalized to the internal control GAPDH using the ΔΔCt method. Primer sequences for SYBR green probes (Integrated DNA Technologies [IDT], Coralville, IA, USA) are the same as used previously [46].

### 4.6. Western Blotting

*Western blotting.* Protein extraction, electrophoresis, and gel transfer to nitrocellulose membranes were performed as previously described [71]. The following primary antibodies were used: rabbit anticleaved-IL-1β, rabbit anti-β-tubulin from Cell Signaling Technology (Danvers, MA). Membranes were either incubated with IRDye 800CW conjugated polyclonal goat antirabbit (1:20,000) from LI-COR Biosciences (Lincoln, NE, USA) and imaged using the Odyssey Infrared Imaging System (LI-COR Biosciences).

### 4.7. Transfection of miRNA, Antagomir and Mimics and siRNA

BEAS-2B cells were transfected with antagomir against miR-23a, miR-23a mimic, Cy3™ dye-labeled pre-miR negative control #1, Cy3-PreCy3 dye-labeled anti-miR-negative inhibitor control (Thermo Fisher, Waltham, MA, USA), siRNA (Dicer-substrate RNAs, DsiRNAs) directed against SIRT3 (IDT), or negative control DsiRNA (IDT) using Lipofectamine 3000 reagent (ThermoFisher) according to the manufacturer’s instructions. Cells were incubated with RNA complexes for 48 h before additional treatment or analysis.

### 4.8. Bacterial Transmigration Assay

Calu-3 cells were grown on 3 μm polyester membrane Costar Transwell^®^ permeable supports (Corning) and tight junctions were mature after a period of 7–10 days. Cells were pretreated with ZLN005 (1 μM) for 18 h. PAO-1 bacterial suspension (MOI of one) was then added apically. After 6 h, the apical and basolateral media were collected separately. A sample of the basolateral medium was serially diluted, cultured on LB agar, and incubated at 37 °C overnight. Colony-forming units were recorded (accurate range, ≥30 ≤ 300), as described previously [71].

### 4.9. Statistical Analysis

All experiments were repeated at least three times. Data are presented as mean ± SEM. Detailed information regarding statistical tests used is included in figure legends. All statistical analyses were performed by using GraphPad Prism (GraphPad Software, La Jolla, CA, USA).

## Figures and Tables

**Figure 1 pathogens-11-00116-f001:**
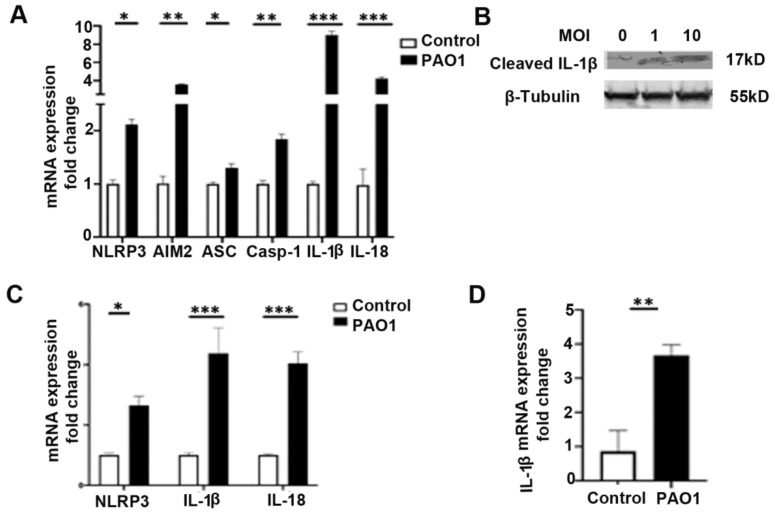
*P. aeruginosa* induces expression of inflammasome components in lung epithelial cells. (**A**) BEAS-2B bronchial epithelial cells were treated with vehicle control (PBS) or infected with the *P. aeruginosa* strain, PAO1, at a multiplicity of infection (MOI) of one for 6 h and then cells were harvested for RNA purification. Relative mRNA expression of nod-like receptor family pyrin domain containing 3 (NLRP3), absent in melanoma 2 (AIM2), apoptosis-associated speck-like protein containing a C-terminal caspase-recruitment domain (ASC), Caspase-1 (Casp-1), IL-1β, and IL-18 were measured. N = 5. (**B**) Western blot for cleaved IL-1β in BEAS-2B bronchial epithelial cells infected with the *P. aeruginosa* strain, PAO1, at an MOI of one and ten for 6 h. (**C**) Primary isolated human bronchial epithelial cells were similarly treated with vehicle control (PBS) or infected with PAO1 at an MOI of one for 6 h. Relative mRNA expression of NLRP3, IL-1β, and IL-18 were measured. N = 5. (**D**) Male C57BL/six mice, 8 to 12 weeks of age, were infected with PAO1 intranasally with 10^8^ CFU or were inoculated with PBS vehicle control. Mice were sacrificed 24 h after intranasal inoculation and the left lung was homogenized and used for mRNA isolation and qPCR to measure relative expression of IL-1β. N = 6. PAO1 infection induces expression of the inflammasome in a bronchial epithelial cell line, primary bronchial epithelial cells, and in vivo in the lungs of mice. Results are mean ± s.e.m. * *p* < 0.05, ** *p* < 0.01, *** *p* < 0.001 all vs. control. Unpaired *t*-tests were used for statistical analysis.

**Figure 2 pathogens-11-00116-f002:**
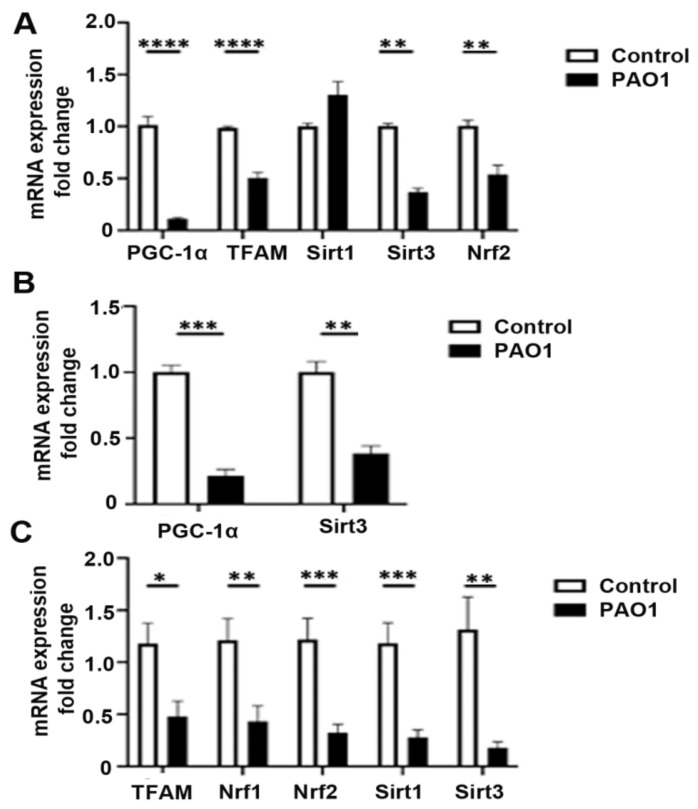
*P. aeruginosa induces expression of inflammasome components in lung epithelial cells.* (**A**) BEAS-2B cells were treated with vehicle control (PBS) or infected with the *P. aeruginosa* strain, PAO1, at a multiplicity of infection (MOI) of one for 6 h and then cells were harvested for RNA purification. Relative mRNA expression of peroxisome proliferator-activated receptor-γ coactivator-1α (PGC-1α), mitochondrial transcription factor A (TFAM), sirtuin 1 (SIRT1), sirtuin 3 (SIRT3), and nuclear factor erythroid 2-related factor 2 (NRF2) were quantified using QPCR. N = 5. (**B**) Primary isolated human bronchial epithelial cells were similarly treated with vehicle control (PBS) or infected with PAO1 at an MOI of one for 6 h. Relative mRNA expression of PGC-1α and SIRT3 were quantified. N = 5. (**C**) Male C57BL/six mice, 8 to 12 weeks of age, were infected with PAO1 intranasally with 10^8^ CFU or were intranasally inoculated with PBS vehicle control. Relative mRNA expression of TFAM, SIRT1, SIRT3, NRF1, and NRF2 were measured from left lung homogenates. N = 6. PAO1 infection attenuates expression of several genes involved in mitochondrial quality control both in vitro in bronchial epithelial cells and in the lungs in a murine pneumonia model. Results are mean ± s.e.m. * *p* < 0.05, ** *p* < 0.01, *** *p* < 0.001, **** *p* < 0.0001 all vs. control. Unpaired *t*-tests were used for statistical analysis.

**Figure 3 pathogens-11-00116-f003:**
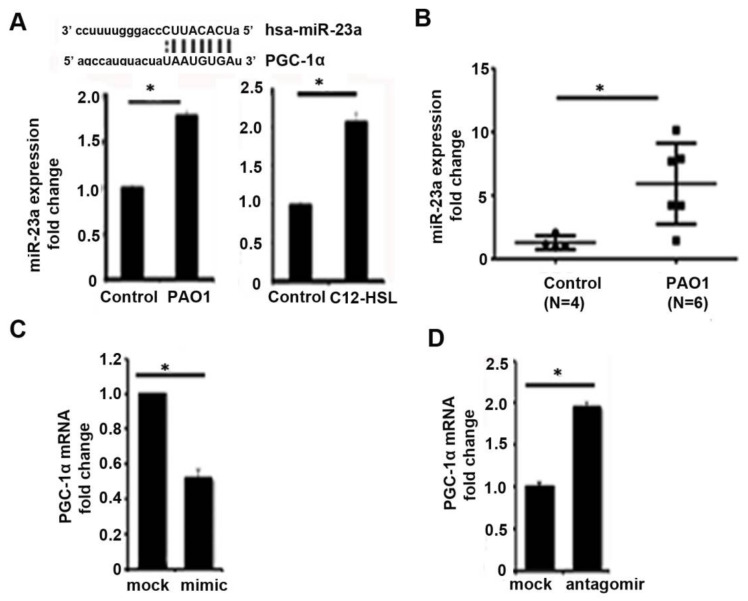
*MiR-23a* acts to post-transcriptionally regulate PGC-1α in *P. aeruginosa* infection. (**A**) BEAS-2B bronchial epithelial cells were treated with vehicle control (PBS) or infected with the *P. aeruginosa* strain, PAO1, at an MOI of one for 6 h or were treated with the *P. aeruginosa* quorum-sensing (QS) molecule, *N*-(3-Oxododecanoyl)-l-homoserine lactone (C12-HSL), at a concentration of 100 μM for 6 h. MicroRNA (miR) were isolated and the relative expression of miR-23a was measured. N = 3. (**B**) Male C57BL/six mice, 8 to 12 weeks of age, were infected with PAO1 intranasally with 10^8^ CFU or were intranasally inoculated with PBS vehicle control. Relative mRNA expression of mir-23a was quantified. N = 4–6. (**C**,**D**) BEAS-2B cells were transfected with mir-23a mimic (**C**) or mir-23a antagomir (**D**) and relative expression of PGC-1α was measured. N = 3. *P. aeruginosa* infection induces mir-23a expression in bronchial epithelial cells and in lung homogenates and mir-23a acts to post-transcriptionally regulate PGC-1α in bronchial epithelial cells. Results are mean ± s.e.m. * *p* < 0.05 vs. control. Unpaired *t*-tests were used for statistical analysis.

**Figure 4 pathogens-11-00116-f004:**
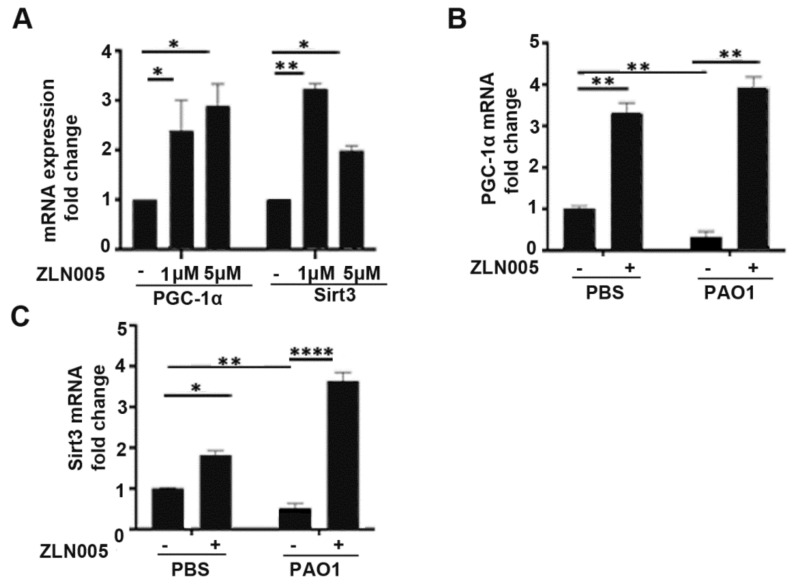
ZLN005 rescue expression of PGC-1α and SIRT3 attenuated by *P. aeruginosa* infection. (**A**) BEAS-2B cells were treated with vehicle control (DMSO) or ZLN005 (1 μM, 5 μM) for 24 h and then relative expression of PGC-1α and SIRT3 was quantified. N = 4. (**B**,**C**) BEAS-2B were pretreated with vehicle control (DMSO) or ZLN005 (1 μM) for 18 h and then cells were infected with PAO1 (MOI 1) or treated with vehicle control (PBS) for an additional 6 h before harvesting. Relative mRNA expression of PGC-1α (**B**) and SIRT3 (**C**) was measured. N = 4. ZLN005 induces expression of PGC-1α and SIRT3 in lung epithelial cells and reverses the down-regulation of PGC-1α and SIRT3 expression caused by *P. aeruginosa* infection. Results are mean ± s.e.m. * *p* < 0.05, ** *p* < 0.01, **** *p* < 0.0001 all vs. control by one-way ANOVA with Tukey’s multiple comparisons test.

**Figure 5 pathogens-11-00116-f005:**
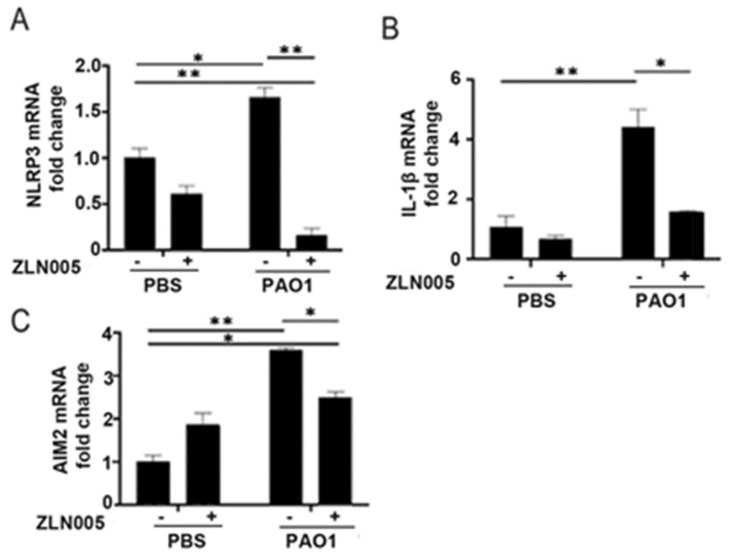
ZLN005 abrogates inflammasome signaling induced by *P. aeruginosa*. (**A**–**C**) BEAS-2B were pretreated with vehicle control (DMSO) or ZLN005 (1 μM) for 18 h and then cells were infected with PAO1 (MOI 1) or treated with vehicle control (PBS) for an additional 6 h before harvesting. Relative mRNA expression of NLRP3 (**A**), IL-1β (**B**), and AIM2 (**C**) were quantified. N = 4. ZLN005 attenuates the upregulation of the NLRP3 and AIM2 inflammasome induced by *P. aeruginosa* infection. Results are mean ± s.e.m. * *p* < 0.05, ** *p* < 0.01 all vs. control by one-way ANOVA with Tukey’s multiple comparisons test.

**Figure 6 pathogens-11-00116-f006:**
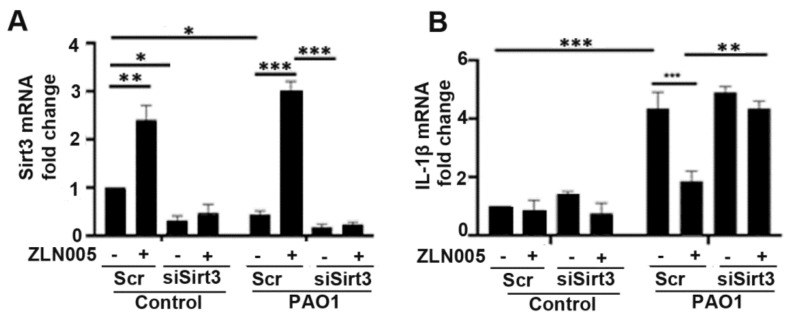
ZLN005 attenuation of inflammasome activation is mediated by activation of the PGC-1α-SIRT3 axis. (**A**,**B**) BEAS-2B were transfected with siRNA directed against sirtuin 3 (siSIRT3) or scrambled RNA (Scr). After 40 h, they were treated with vehicle control (DMSO) or ZLN005 (1 μM) for 18 h prior to inoculation with PAO1 (MOI 1) or PBS for an additional 6 h. Relative mRNA expression of SIRT3 and IL-1β was then measured. N = 5. Knockdown of SIRT3 expression abrogates the attenuation of inflammasome signaling caused by ZLN005 treatment. Results are mean ± s.e.m. * *p* < 0.05, ** *p* < 0.01, *** *p* < 0.001 all vs. control by one-way ANOVA with Tukey’s multiple comparisons test.

**Figure 7 pathogens-11-00116-f007:**
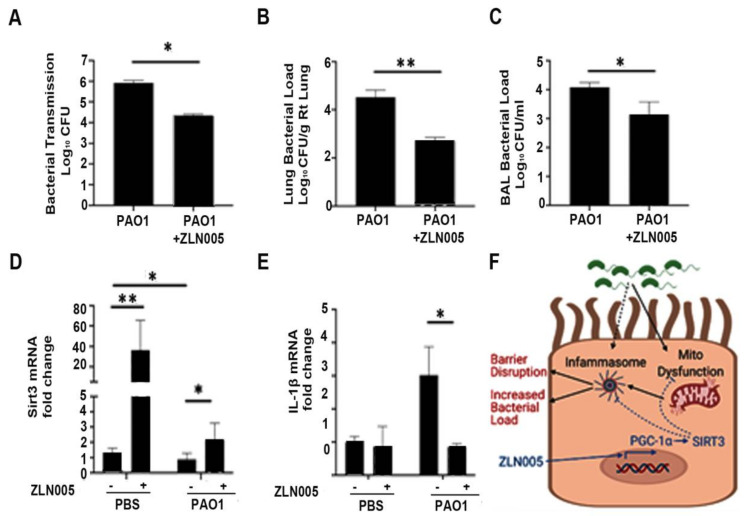
ZLN005 improves epithelial barrier function and reduces bacterial load in vivo. (**A**) Calu-3 lung epithelial cells were grown on transwell supports for 7–10 days to allow for tight junctions to mature. Cells were then treated with ZLN005 (1 μM) in the apical and basolateral media. After 18 h, the cells were treated with vehicle control (PBS) or PAO1 (MOI 1) in the apical chamber. After 6 h, the basolateral media was collected and serially diluted and cultured on LB agar. Colony-forming units (CFU) were recorded. N = 6. (**B**–**E**) Male C57BL/six mice, aged 8 to 12 weeks, were treated with ZLN005 (15 mg/kg body weight) or with an equivalent volume of vehicle control (methylcellulose) for three doses every 8 h for one day. At the time of the last dose, the mice were then intranasally inoculated with vehicle control (PBS) or with PAO1 (10^8^ CFU). Mice were sacrificed after 24 h. Bronchoalveolar lavage (BAL) fluid was collected and serially diluted and used for bacterial colony counts (**D**). The right lung was collected, weighed, homogenized, and serially diluted on LB agar plates to quantify bacterial load normalized to the weight of the lung (**C**). The left lung was frozen and used for subsequent mRNA analyses to measure mRNA levels of SIRT3 and IL-1β (**D**,**E**). N = 3–6. ZLN005 attenuated bacterial transmigration across the epithelial barrier. Further, ZLN005 reduced bacterial load in an in vivo *P. aeruginosa* pneumonia model, rescued expression of SIRT3, and reduced IL-1β expression in whole lungs. Results are mean ± s.e.m. * *p* < 0.05, ** *p* < 0.01 vs. control by one-way ANOVA with Tukey’s multiple comparisons test. (**F**) Schematic representation of the effect of ZLN005 on attenuating mitochondrial dysfunction and inflammasome activation via induction of the PGC-1α-SIRT3 axis. Figure was created with BioRender.com (accessed on 30 October 2021).

## Data Availability

All data generated or analyzed during this study are included in this published article.

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
