# Peer review of "The Effect of PGC-1alpha-SIRT3 Pathway Activation on Pseudomonas aeruginosa Infection"

_pathogens, 2022, doi:10.3390/pathogens11020116_

Round 1

Reviewer 1 Report

Major comments:
1) In most of this study, including animal experiments, the authors observed the effect of ZLN005, an activator of PGC-1α, on Pseudomonas aeruginosa infection, although there are some experiments on SIRT3 by miR-23a mimic and anti-miR-23a.
This study itself proceeded on the premise that the peroxisome proliferator-activated receptor-γ coactivator-1α inhibitor ZLN005 suppresses inflammasome formation. However, the authors did not observe inflammasome formation itself. The title of this paper should be "the effect of peroxisome proliferator-activated receptor-γ coactivator-1α inhibitor ZLN005 on Pseudomonas aeruginosa infection", or something a mixutre of "ZLN005" with  SIRT3 pathway.

2) It is strange that there is no description about ZLN005 in the introduction. In the final section of the introduction, it should be noted that this experiment saw the effect of ZLN005 on Pseudomonas aeruginosa infection.

3) Some limitations of this study should also be summarized in the discussion. This study uses PAO1, a representative strain of Pseudomonas aeruginosa. In Pseudomonas aeruginosa, the PAO1 strain was the first target for whole-genome analysis and is undoubtedly a standard laboratory strain. However, although the PAO1 strain exhibits intermediate toxicity among Pseudomonas aeruginosa strains, it is neither a clinically problematic mucoid strain nor an exoU-positive strain with strong cytotoxicity. Therefore, the results obtained by the authors are only from one representative strain, and the results may change if authors use a different Pseudomonas aeruginosa strain. 

4) The importance of the authors' findings is probably of great interest in their potential as a therapeutic agent for cystic fibrosis patients who are chronically infected with Pseudomonas aeruginosa mucoid strains and have chronic inflammatory conditions. I want the authors to add comments about that to the discussion.  In addition, in the clinical practice of critical care, the main problem is an infection caused by multidrug-resistant Pseudomonas aeruginosa in immunocompromised patients. The effect on pattern recognition receptors such as PAMPS and DAMPS cannot be easily reduced in the site of infection due to the dead pathogens and destructed cell components even if the antibacterial agent is effectively used. Therefore, the authors' idea of using ZLN005, or ZLN005-like molecule as an adjuvant of antibacterials may be effective in the control of the inflammation induced by bacterial infections. The possible advantages of the evidence obtained from this study, such as the combination of antibacterial and anti-inflammatory agents in patients with chronic infections and PGC-1α activator, should be discussed more in Discussion.

Minor Comments:
1) 
The authors are experimenting with MOI = 1 in in-vitro experiments, but what if a higher MOI is used?
2) p5, line 1 70: Typo error: In bronchial epithelial cells (B[extra space] EAS-2B).

Author Response

We thank the reviewers for their thoughtful reviews of our manuscript and insightful comments to address major and minor concerns of the paper. We believe that because of the changes we have made to the manuscript highlighted below in bold, we have significantly improved the quality of the paper.

Major comments:
1) In most of this study, including animal experiments, the authors observed the effect of ZLN005, an activator of PGC-1α, on Pseudomonas aeruginosa infection, although there are some experiments on SIRT3 by miR-23a mimic and anti-miR-23a.
This study itself proceeded on the premise that the peroxisome proliferator-activated receptor-γ coactivator-1α inhibitor ZLN005 suppresses inflammasome formation. However, the authors did not observe inflammasome formation itself. The title of this paper should be "the effect of peroxisome proliferator-activated receptor-γ coactivator-1α inhibitor ZLN005 on Pseudomonas aeruginosa infection", or something a mixutre of "ZLN005" with  SIRT3 pathway.

The title has been changed to “The effect of PGC-1alpha-SIRT3 pathway activation on Pseudomonas aeruginosa infection”

2) It is strange that there is no description about ZLN005 in the introduction. In the final section of the introduction, it should be noted that this experiment saw the effect of ZLN005 on Pseudomonas aeruginosa infection.

This has been added to the Introduction.

3) Some limitations of this study should also be summarized in the discussion. This study uses PAO1, a representative strain of Pseudomonas aeruginosa. In Pseudomonas aeruginosa, the PAO1 strain was the first target for whole-genome analysis and is undoubtedly a standard laboratory strain. However, although the PAO1 strain exhibits intermediate toxicity among Pseudomonas aeruginosa strains, it is neither a clinically problematic mucoid strain nor an exoU-positive strain with strong cytotoxicity. Therefore, the results obtained by the authors are only from one representative strain, and the results may change if authors use a different Pseudomonas aeruginosa strain. 

This important limitation has been incorporated in the Discussion.

4) The importance of the authors' findings is probably of great interest in their potential as a therapeutic agent for cystic fibrosis patients who are chronically infected with Pseudomonas aeruginosa mucoid strains and have chronic inflammatory conditions. I want the authors to add comments about that to the discussion.  In addition, in the clinical practice of critical care, the main problem is an infection caused by multidrug-resistant Pseudomonas aeruginosa in immunocompromised patients. The effect on pattern recognition receptors such as PAMPS and DAMPS cannot be easily reduced in the site of infection due to the dead pathogens and destructed cell components even if the antibacterial agent is effectively used. Therefore, the authors' idea of using ZLN005, or ZLN005-like molecule as an adjuvant of antibacterials may be effective in the control of the inflammation induced by bacterial infections. The possible advantages of the evidence obtained from this study, such as the combination of antibacterial and anti-inflammatory agents in patients with chronic infections and PGC-1α activator, should be discussed more in Discussion.

This has been added to the Discussion.

Minor Comments:
1) 
The authors are experimenting with MOI = 1 in in-vitro experiments, but what if a higher MOI is used?

A multiplicity of infection (MOI) of 1 was chosen based on optimization experiments that found high toxicity with higher doses (MOI 10-30). See Page 4 Line 150

2) p5, line 1 70: Typo error: In bronchial epithelial cells (B[extra space] EAS-2B).

This has been corrected.

Reviewer 2 Report

In the article “Pseudomonas aeruginosa inflammasome activation attenuated by the PGC-1alpha-SIRT3 pathway” the authors investigate the role of ZLN0005 compound in the attenuation of inflammasome signalling induced by Pseudomonas aeruginosa pulmonary infection. In particular, the investigated pathway is the PGC-1alpha-SIRT3 axis, which upstream regulates the mitochondrial biogenesis, cellular respiration and antioxidant defence and downstream is involved in cellular metabolic processes and reactive oxygen species homeostasis.

The rationale behind the paper is clear and the manuscript is overall well written and easy to follow and comprehend. There are anyway few concerns to be addressed.

First, the study evaluates the effect of the P. aeruginosa infection showing a reduction in the expression levels of several inflammasome components in vitro, in vivo. The only concern is that in the paragraph 2.1. the authors focus only on the evaluation of mRNA levels of the target proteins, while it would be interesting an evaluation also of the protein levels, since there might be some post-transcriptional regulation also of those genes as shown later. Moreover, IL-1beta levels can be also assessed by ELISA tests. Concerning IL-1beta, is confusing why sometimes the authors check for IL1beta expression and sometimes for the pro-IL-1beta (see paragraph 2.5). Secondly, why in paragraph 2.2. the analysed genes are different between all the 3 conditions (namely, epithelial cells, primary cells and lungs)?

Then, in figure 3C and 3D: which is the n of these experiments? The mock bars look like is just N=1.  

Minor:

The graphs of all the figures have different styles, please apply an uniform style.

Line 62: space typo

Line 68: LPS and flagellin

Line 68-70: This sentence is not clear, please rephrase.

Line 73: Calcium release is not a DAMPs.

Line 75-77: The sentence is not clear please rephrase.

Figure 1C: parenthesis for the FOLD CHANGE

Figure 2A and 2B: parenthesis for the FOLD CHANGE

Line 142: MOI abbreviation already explained

Figure 2: SIRT1 and Nrf1 are not mentioned but used for the experiment.

Line 170: typing mistake in BEAS-2B (space between b and eas)

Figure 3A, 3B, 3C: parenthesis for the FOLD CHANGE and please use the same style for the graphs.

Line 387: paragraph 4.3. does not exist.

The reference 68 cited in the materials and methods section do not contain the named protocol.

Author Response

Comments and Suggestions for Authors

In the article “Pseudomonas aeruginosa inflammasome activation attenuated by the PGC-1alpha-SIRT3 pathway” the authors investigate the role of ZLN0005 compound in the attenuation of inflammasome signalling induced by Pseudomonas aeruginosa pulmonary infection. In particular, the investigated pathway is the PGC-1alpha-SIRT3 axis, which upstream regulates the mitochondrial biogenesis, cellular respiration and antioxidant defence and downstream is involved in cellular metabolic processes and reactive oxygen species homeostasis.

The rationale behind the paper is clear and the manuscript is overall well written and easy to follow and comprehend. There are anyway few concerns to be addressed.

First, the study evaluates the effect of the P. aeruginosa infection showing a reduction in the expression levels of several inflammasome components in vitro, in vivo. The only concern is that in the paragraph 2.1. the authors focus only on the evaluation of mRNA levels of the target proteins, while it would be interesting an evaluation also of the protein levels, since there might be some post-transcriptional regulation also of those genes as shown later. Moreover, IL-1beta levels can be also assessed by ELISA tests. Concerning IL-1beta, is confusing why sometimes the authors check for IL1beta expression and sometimes for the pro-IL-1beta (see paragraph 2.5).

We have added Fig 1B. which demonstrates that P. aeruginosa infection increases post-transcriptional processing of IL-1 beta. The term pro-IL-1 beta was used in certain places in the manuscript because this is an alternative gene name for the IL-1 beta gene. It has been removed for simplicity since IL-1 beta is the official gene name.

Secondly, why in paragraph 2.2. the analysed genes are different between all the 3 conditions (namely, epithelial cells, primary cells and lungs)?

Non-significant changes in NRF1 in epithelial cells and PGC-1alpha in vivo were not included in original submission but this has been added to paragraph 2.2. Because of limited mRNA available, we only analyzed PGC1-alpha and SIRT3 in primary human cells since this axis was found to be important in later experiments.

Then, in figure 3C and 3D: which is the n of these experiments? The mock bars look like is just N=1.  

N =3 for Fig 3C and 3D. Caption has been updated.

Minor:

The graphs of all the figures have different styles, please apply an uniform style.

Control experiments are white bars. PAO1 infection groups are black bars. ZLN treatment is indicated with diagonal bars.

Line 62: space typo

This has been corrected

Line 68: LPS and flagellin

This has been corrected.

Line 68-70: This sentence is not clear, please rephrase.

This has been rephrased.

Line 73: Calcium release is not a DAMPs.

This has been deleted.

Line 75-77: The sentence is not clear please rephrase.

This has been rephrased.

Figure 1C: parenthesis for the FOLD CHANGE

Parentheses have been deleted.

Figure 2A and 2B: parenthesis for the FOLD CHANGE

Parentheses have been deleted.

Line 142: MOI abbreviation already explained

This has been corrected.

Figure 2: SIRT1 and Nrf1 are not mentioned but used for the experiment.

This has been corrected.

Line 170: typing mistake in BEAS-2B (space between b and eas)

This has been corrected.

Figure 3A, 3B, 3C: parenthesis for the FOLD CHANGE and please use the same style for the graphs.

This has been corrected.

Line 387: paragraph 4.3. does not exist.

This has been corrected.

The reference 68 cited in the materials and methods section do not contain the named protocol.

The reference has been replaced with the correct reference

Round 2

Reviewer 2 Report

I would like to thank the authors for their efforts in replying to all the comments provided.

I have still a concern regarding the style of the graphs, since are used different styles among all the figures. In my opinion, the error bars, the statistical representation, the font suite and the font type have to be the same throughout the figures. 

Author Response

Thanks for comments.

We re-edit all of figures in the manuscript.